# Antibiotic Use in Alpine Dairy Farms and Its Relation to Biosecurity and Animal Welfare

**DOI:** 10.3390/antibiotics11020231

**Published:** 2022-02-10

**Authors:** Francesca Menegon, Katia Capello, Jacopo Tarakdjian, Dario Pasqualin, Giovanni Cunial, Sara Andreatta, Debora Dellamaria, Grazia Manca, Giovanni Farina, Guido Di Martino

**Affiliations:** Istituto Zooprofilattico Sperimentale delle Venezie, Viale dell’Università 10, 35020 Legnaro, Italy; fmenegon@izsvenezie.it (F.M.); kcapello@izsvenezie.it (K.C.); jtarakdjian@izsvenezie.it (J.T.); dpasqualin@izsvenezie.it (D.P.); gcunial@izsvenezie.it (G.C.); sandreatta@izsvenezie.it (S.A.); ddellamaria@izsvenezie.it (D.D.); gmanca@izsvenezie.it (G.M.); gfarina@izsvenezie.it (G.F.)

**Keywords:** AMU, antibiotics, dairy cattle, DDD, biosecurity, animal welfare

## Abstract

The quantification of antimicrobial usage (AMU) in food-producing animals can help identify AMU risk factors, thereby enhancing appropriate stewardship policies and strategies for a more rational use. AMU in a sample of 34 farms in the Province of Trento (north-eastern Italy) from 2018 to 2020 was expressed as defined daily doses for animals per population correction unit according to European Surveillance of Veterinary Antimicrobial Consumption guidelines (DDDvet) and according to Italian guidelines (DDDAit). A retrospective analysis was carried out to test the effects of several husbandry practices on AMU. Overall, the average AMU ranged between 6.5 DDDAit in 2018 and 5.2 DDDAit in 2020 (corresponding to 9 and 7 DDDvet, respectively), showing a significant trend of decrement (−21.3%). Usage of the highest priority critically important antimicrobials (HPCIA) was reduced by 83% from 2018 to 2020. Quarantine management, available space, water supply, animals’ cleanliness and somatic cell count had no significant association with AMU. Rather, farms with straw-bedded cubicles had lower AMU levels than those with mattresses and concrete floors (*p* < 0.05). In conclusion, this study evidenced a decrement in AMU, particularly regarding HPCIA, but only a few risk factors due to farm management.

## 1. Introduction

The usage of antimicrobials (AMU) to treat large animal populations in intensive livestock production is necessary to avoid animals suffering from bacterial infections and, therefore, to guarantee animal health and welfare [1], but it comes with side effects. In fact, antimicrobial use in both humans and animals leads to antimicrobial resistance (AMR) [2,3,4,5].

AMR is a global threat for both human and animal health, as it might compromise the effectiveness of infections’ treatment [4,6,7]. In particular, misuse, underdosing and overuse of antimicrobials are the strongest drivers for AMR [8,9]. Hence, action plans on AMR were developed both at a national and international level [9,10,11]. At the national level, the Italian Ministry of Health promoted a plan against AMR [12], calling for antimicrobial stewardship, with the ultimate goal of providing a coordinated and sustainable strategy to address AMR nationwide. For this purpose, possible benchmarks for farm categorization according to sanitary risk, animal welfare and antimicrobial consumption have been identified [13]. This approach is also endorsed by the EU Farm to Fork strategy [14], which supports a better welfare to improve animal health and food quality and also reduce the need for antimicrobials [15,16]. Within this framework, biosecurity includes all procedures to prevent pathogens from entering a farm and spreading within a farm [17,18]. An improved biosecurity level was proven to positively impact animal health and welfare [17] and to reduce AMU in pigs [19,20,21] and beef cattle [22]. For dairy cattle, various studies in the literature investigated AMU in cows and calves [23,24,25,26]. However, only limited information is available on possible associations between AMU, biosecurity and animal welfare. There are only a few studies available on AMU in the Italian dairy sector, and they do not cover the large variety of livestock systems [27,28]. Therefore, the present study aimed at quantifying AMU in a geographically defined dairy cattle population in Northern Italy (Trentino Alto Adige region) for three years (2018–2020), applying two different dose-based methods. AMU associations with several management factors of biosecurity and animal welfare were also investigated.

## 2. Results

The average amount of AMU in the inspected 34 dairy farms in 2018, 2019, 2020 was, respectively, 6.48, 6.58 and 5.18 DDDAit/PCU, with a significant decrease (*p* < 0.001) in 2020 compared to both 2018 (−20.1%) and 2019 (−21.3%). Regarding AMU using DDDvet/PCU, the observed average values were 9.03, 8.60, 6.94 for 2018, 2019 and 2020, respectively. Penicillins and first and second generation cephalosporins were the most frequently administered antimicrobial classes, accounting for 34.85% and 13.05% of the overall AMU (Table 1). Antimicrobials classified as highest priority critically important (HPCIA), ranged between 23.52% (in 2018) and 5.32% (in 2020) DDDAit of the total AMU (Table 1). The use of HPCIA was reduced by 83% from 2018 to 2020 (Table 1). HPCIA were more frequently administered via injectable products (Table 2).

Intramammary treatment was the most frequent administration route, representing 53.11% of the total AMU, followed by injectable (45.62%), intrauterine (1.18%) and oral administration routes (0.09%). A significantly lower mean value of DDDAit was observed for all years in lactation compared to dry-off (Table 2), representing, respectively, 37.71% and 62.28% of the intramammary products used.

The analysis of management factors on AMU is shown in Table 3. Five farms performed blanket DCT (dry cow therapy) (3.36 ± 0.22 DDDAit), while 29 farms performed selective DCT using on average 1.44 ± 0.65 DDDAit during the dry-off period. Moreover, in the majority of the farms (i.e., 28 out of 34), microbiological tests for mastitis were routinely performed in symptomatic animals. The presence of a quarantine box, space availability, water quality, animals’ cleanliness, presence of ventilation alarm and level of somatic cell count (SCC) did not exhibit any significant association with AMU. Instead, a lower DDDAit was found (*p* = 0.050) in straw/sawdust bedded cubicles compared to other materials (i.e., mattresses and concrete floor). Overall, the average biosecurity score recorded in farms was 65% (range 37–85%), while the animal welfare average score was 77% (range 63–88%); nonetheless, no significant correlation was found with AMU distribution. On the contrary, the quantity of daily milk produced (mean value 29.7 ± 5.44) was positively correlated with the DDDAit (Spearman’s rho = 0.37, *p* = 0.027).

## 3. Discussion

This study evaluated AMU retrospectively in a sample of holdings, mostly family-run, that we can assume are representative of Alpine dairy farming systems, characterized by small–medium size, usually with access to pasture and without a milking robot. As reported in the European Surveillance of Veterinary Antimicrobial Consumption (ESVAC) in the last decade, Italy was ranked high in Europe in terms of AMU [29]. Compared to other EU studies, this occurrence seems to be confirmed by the present data. For example, an English study [23] reported an average DDDvet of 4.60. However, comparison should be made with caution because these authors did not consider DCT. In terms of DDDAit, the dose-based values in the present study were lower than for DDDvet. DDDAit was defined for each active substance following the summary of Italian products’ characteristics: this guaranteed a higher level of precision than DDDvet. Average DDDAit ranged yearly between 6.58 in 2019 and 5.18 in 2020, with a significant variability between farms (2.19–10.34). Considering a three-year timeframe can overcome outlier peak values due to exceptional circumstances and evidenced a significant decreasing trend. A previous Italian study [30] assessed AMU in dairy cattle farms using DDDAit in 2019 in north-western Italy (Lombardy region). These authors observed a similar AMU compared to the present study (4.8 DDDAit per cow/year). However, a different methodology calculation in intramammary products was applied. The number of doses was divided by the Population Correction Unit (PCU) (likewise injectable products) and not by the number of cows, as proposed by the Italian Ministry of Health for this route of administration [31]. The chosen approach can generate an apparently lower consumption estimation compared to the present study. In line with the massive decrease in sales of veterinary antibiotics in European Countries, highlighted in the 10th ESVAC report, the results of the present study show a reduction of 21.3% from 2019 to 2020. This decreasing trend had already been observed from 2007 to 2012 in the Netherlands, where dairy farms registered a −22% reduction of AMU [32]. Conversely, Mazza et al. did not observe a significant decrement in Lombardy. Furthermore, a more prudent AMU was observed during the study period in Alpine farms, with a dramatic reduction in HPCIA (−83%). They were eliminated in intramammary treatments in nearly all the farms. These data are encouraging, considering that in 2015 in Austria, a mean 1.14 DDDvet of HPCIA was used for lactating cows’ treatments with intramammary products [33]. In US dairy farms in 2016–2017, 75% of intramammary antimicrobials contained HPCIAs such as ceftiofur and 7% contained cephapirin. These two active principles accounted for almost 82% of the total AMU [34]. The difference between farming system types can also play a major role. Indeed, it has been recently shown [35] that mountain farms with smaller herd sizes, which provide cows with access to pasture and limit concentrates in the diet, use less HPCIA than specialized dairy farms from lowlands. Moreover, it has been widely reported that higher productivity (notably higher milk yield) can be associated with a higher incidence of metabolic disorders, such as mastitis, lameness and other production diseases [36], requiring prompt veterinary intervention. Our results support this evidence, as we traced a positive correlation between milk production and AMU. In line with what is reported in the literature for dairy cattle [27,30,32,34], we found that intramammary medication is the most frequent administration route (53.11%). Previous studies observed a frequency reaching 63% [32] and 78% [34]. In particular, antimicrobial treatments were reported to be more frequent during the dry-off period than during lactation. Dry-off treatments are mainly administered to prevent mastitis in the following lactation [37] but in the present study blanket DCT was performed only in 5 out of 34 farms in 2020. Indeed, most farms (28/34) performed microbiological tests for mastitis in symptomatic animals.

However, no correlation between AMU and SCC was found in the present study. A previous study demonstrated that it is possible to reduce antibiotics in dry-off without increasing somatic cells count (SCC) in the following lactation by implementing management measures to maintain good udder health [38]. For example, some risk factors for *E. coli* mastitis at the herd level include the bedding material and the design of cubicles [39]. The number of cubicles was classified as acceptable in all inspected farms, according to Italian welfare guidelines [13] (ranged between 90–110% of the number of cows) thus this parameter was not an object of analysis. We grouped bedding materials that needed to be frequently replaced, such as straw, sawdust or both, and we compared them to other materials (including mattresses and concrete floor). Therefore, farms that used replaceable materials had lower AMU, which can be attributed to a higher hygiene level guaranteed by a regular replacement of new material. Moreover, in the past, straw also showed a lower incidence of leg injuries in the tarsal joints [40,41,42], in addition to promoting better animal welfare resulting from higher comfort [42,43] and the presence of manipulable material. On the other hand, other studies observed that sawdust increases the dirtiness of the hindquarters and udder [44] and, compared to straw, it increases the risk of hock skin alterations [45]. Considering these studies, straw can be assumed to be better than sawdust. However, this study does not consider that frequently dry cows are housed separately from lactating cows, and data do not distinguish eventual differences as bedding material between the two phases. Therefore, more detailed studies will be required, including the distinction between dry and lactation period and between straw, sawdust or the use of both of them. No significant correlation between AMU and biosecurity as well as animal welfare scores was observed in the present study. Additionally, no correlation between AMU, biosecurity and animal welfare was found in dairy cattle farms from Lombardy [27], which had similar scores (biosecurity range 21.41–71.56%, animal welfare range 45.32–81.69%). In the beef cattle sector, an improved level of welfare permitted lower usage of antimicrobials, but not a reduction in HPCIA [22]. On the contrary, in 61 Belgian pig farms, AMU was reduced by 52.0%, without impairing the herd production performances, by adopting a series of herd-level interventions concerning biosecurity, vaccination scheme, health care, welfare and zootechnical measures [46]. Similarly, a reduction of 47% of AMU was achieved in pig farms from Belgium, France, Germany and Sweden from birth to slaughter, and farms with bigger compliance achieved a more considerable reduction [19]. However, pig farming is characterized by a general higher AMU than dairy cows (e.g., 16 DDDvet/PCU in 2017 for Tarakdjian et al., 2020 [47]), and this evidence can explain why good management practices are more likely to produce a significant effect in the former than in the latter livestock sector.

## 4. Materials and Methods

### 4.1. Data Collection

Thirty-four free-stall farms of mixed-breed dairy cows (i.e., Holstein Friesian, Brown Swiss, Simmental cows) in the Province of Trento (north-eastern Italy) were randomly selected for this study. Farms were located in the area between Garda Lake (south), Stelvio Natural Park (west), Pale di San Martino Natural Park (east) and Bolzano (north), and the altitude ranged from 400 m to 1700 m. Median farm size was 89.5 animals and the mean was 140, representing a small–medium farm in the Italian dairy sector. In 26 of these farms, animals had access to pasture. Farms were visited in 2021; data were collected retrospectively from the farm registry from 2018 to 2020. The biosecurity score was calculated using the Classyfarm protocol, available in Ginestreti et al. [27], and considers 15 items to compute a final score. The final score ranges from 0% to 100%, where 100% is the highest level of biosecurity. Animal welfare was evaluated using Classyfarm protocol, as previously described in Ginestreti et al. [27] and Mazza et al. [30]. The score includes 70 items, 18 of which are animal-based measures (details available in Bertocchi et al. [48]). The final score ranges from 0 to 100%, where 100% corresponds to the maximum level of animal welfare. The most relevant biosecurity and welfare items were also analyzed separately as qualitative variables. These items were the following: average mortality rate (higher or lower than 5%), presence of ventilation alarm and replacement equipment, presence of sick animals’ boxes, space available, presence of quarantine boxes, animal cleanliness (less or more than 20% of dirty animals), water (at least one drinker/10 animals), access to pasture, bedding material (straw and/or sawdust/others) and presence of laboratory analyses for mastitis (absent, not on a routine basis, for all problematic cows). Moreover, during the inspection, it was possible to obtain data on average yearly and daily milk production and somatic cell count (SCC) (> or ≤150,000 cells/mL). Based on veterinarians’ declarations, it was also possible to distinguish farms applying blanket DCT vs. selective DCT. For each farm, the average number of animals present in the farm for one year was calculated by performing three four-monthly extractions per year from the National data Registry.

### 4.2. Data Analysis

Overall, AMU per year was quantified using the Defined Daily Dose/Population Cor-rection Unit (DDDAit/PCU) method, proposed by the Italian Ministry of Health [31]. The total amount of mg of an active substance administered for each commercial product was divided by the dosage reported in the summary of products characteristics (SPC), which was reported in the official pharmaceutical handbook [49]. To allow a broader comparison with studies performed in other European countries, AMU was also quantified using DDDvet/PCU [50]. The European Medicines Agency (EMA) provides a standardized daily dosage for each active substance, calculated as the average concentration of various marketed products sold in the European Union (EU) [50]. All intramammary, injectable, intrauterine and oral products were analyzed. Spray AM for external use and ruminal preparations (monensin) were excluded. Oral and injectable antibiotic formulations were calculated per kilogram of animal, while intrauterine and lactating cow intramammary doses (IMM) were calculated per cow. If a range of doses was given, the mean dose was used as the DDDAit [31]. For combination products (e.g., penicillin-streptomycin), the DDDvet was estimated for the main substance, following the guidelines for the DDDvet assignment [50].

Conversely, DDDAit was estimated for both separately, according to the Italian Ministry of Health guidelines [31]. The “PCU” (Population Correction Unit), representing the entire animal biomass “at-risk” present annually in the farm, was calculated by dividing the animals into production categories and multiplying the results by the standard weight attributed to each category. According to the National Reference Centre for Animal Welfare (CreNBA) [31] guidelines, calf, heifer and cow weights were attributed as 100 kg, 300 kg and 600 kg, respectively. Conversely, according to EMA guidelines for DDDvet, the weights attributed were 140 kg, 200 kg, 425 kg for the same previous livestock categories [50]. Antibiotics were grouped according to EMA and WHO classifications, based on their importance to human health [51,52].

### 4.3. Statistical Analysis

The effects of the phase (dry-off/lactation), year and their interaction on DDDAit distribution were assessed using a linear mixed model; the farm was considered in the model as random effect while the type of phase and year within each farm was included in the repeated effects part of the model, using a compound symmetry covariance structure. The Akaike Information Criterion (AIC) and the residual diagnostics were used to evaluate the model’s goodness of fit. Given the model results, the evaluation of the factors collected for each farm in comparison with the DDDAit values was performed considering the year 2020. The Wilcoxon rank-sum test was applied in the case of categorical variables, after having checked the equality of variances using Leven’s robust test statistic; regarding continuous variables (i.e., daily milk production, biosecurity and animal welfare scores), the Spearman correlation coefficient was adopted. The HPCIA usage evaluation was performed after dichotomizing the variable (i.e., used at least once or not); given the observed frequencies, only the comparison between 2019 and 2020 for the injectable products was statistically analyzed, using the McNemar test. Given the massive reduction in 2020 of the HPCIAs (only 18 farms used HPCIAs at least once) and the consequent sample size reduction after stratifying by the management variables, HPCIA were not included in the risk factors analysis since the robustness of the statistical analysis would not have been satisfactory. Data analysis was conducted using software SAS v.9.4 (SAS Institute Inc., Cary, NC, USA).

## 5. Conclusions

This study evidenced a significant decrement in AMU in 34 Alpine dairy farms in 2018–2020, with particular reference to HPCIA. Penicillins and first and second generation cephalosporins were the most frequently administered antimicrobial classes and the most widely used administration route was the intramammary one, specifically during the dry-off period. Replaceable cubicle materials were associated with lower AMU, while quarantine management, available space, water supply, animals’ cleanliness and somatic cell count had no significant association with AMU. Furthermore, no association between overall AW and biosecurity score was found. If confirmed by further studies, these results may suggest benchmarks for Alpine dairy farms to promote the reduction of AMU in dairy industry.

## Figures and Tables

**Table 1 antibiotics-11-00231-t001:** Percentages of defined daily doses for animals per population correction unit (DDDAit/PCU and DDDvet/PCU) administered in 2018–2020 in 34 Alpine dairy cattle farms: Antimicrobials have been grouped and classified according to WHO categorization: HPCIA: highest priority critically important antimicrobials; CIA: critically important antimicrobials; HIA: highly important antimicrobials; IA: important antimicrobials.

	DDDAit %			DDDvet %		
	2018	2019	2020	2018	2019	2020
HPCIA	23.52	11.03	5.32	27.50	13.26	6.05
Cephalosporins 3rd gen.	10.40	7.05	3.49	14.42	9.14	4.52
Cephalosporins 4th gen.	3.76	0.61	0.00	5.18	0.21	0.00
Fluoroquinolones	6.74	2.09	1.00	5.98	2.19	0.97
Macrolides	2.62	1.29	0.83	1.92	1.72	0.56
Polymyxins	0.00	0.00	0.00	0.01	0.00	0.00
CIA	45.41	47.12	47.55	42.15	46.27	49.91
Aminoglycosides	7.18	7.40	7.32	4.95	5.55	6.06
Ansamycin	5.34	3.55	4.50	6.72	5.38	5.47
Penicillins	32.90	36.16	35.72	30.49	35.35	38.38
HIA	25.01	35.72	36.69	26.55	36.64	37.50
Amphenicols	0.22	0.14	0.12	0.43	0.26	0.22
Cephalosporins 1st and 2nd gen.	11.26	15.81	11.91	14.39	17.65	14.81
Lincosamides	6.09	6.18	10.18	4.13	4.19	7.03
Sulphonamides	3.82	9.74	9.92	3.62	9.94	10.18
Tetracyclines	3.62	3.84	4.56	3.97	4.60	5.26
IA	6.17	6.14	10.44	3.79	3.82	6.54
Aminocyclitols	6.17	6.14	10.44	3.79	3.82	6.54

**Table 2 antibiotics-11-00231-t002:** Average Defined daily doses for animals per population correction unit (DDDAit/PCU) administered in 2018–2020 in 34 Alpine dairy cattle farms: Intramammary administered antimicrobials are divided into “dry-off” and “lactation” products, while injectable, intrauterine and oral products are grouped in “not intramammary product” (not IMM). The percentage of highest priority critically important antimicrobials (HPCIA) is also reported.

		2018		2019		2020		Year (Y)	Phase (P)	Y × F
Phase	n.	Mean ± sd	% HPCIA	Mean ± sd	%HPCIA	Mean ± sd	% HPCIA			
dry-off	34	2.21 ± 1.21	20.59%	2.45 ± 1.73	5.88%	1.51 ± 1.08	2.94%	<0.001	<0.001	0.14
lactation	34	1.42 ± 1.11	35.29%	1.21 ± 0.96	14.71%	1.05 ± 0.80	5.88%			
not IMM	34	3.10 ± 1.68	94.12%	2.91 ± 1.81	76.47%	2.74 ± 1.96	52.94%			

**Table 3 antibiotics-11-00231-t003:** Associations between defined daily doses per animal (DDDAit/PCU) and different management variables in 34 Alpine dairy cattle farms in 2020. Significance for *p* < 0.05.

	n	Mean ± sd	*p* Value
Quarantine box			
absent	13	5.87 ± 3.25	ns
present	21	4.95 ± 2.19	
Mortality			
<5%	27	5.42 ± 2.76	na
≥5%	7	4.85 ± 2.18	
Sickbay			
present	31	5.14 ± 2.70	na
absent	3	6.98 ± 0.85	
Space available (heifers)			
≥3.5 m^2^/animal	20	5.35 ± 2.21	ns
<3.5 m^2^/animal	14	5.23 ± 3.24	
Water supply			
≥1 drinker/10 animals	19	5.08 ± 2.06	ns
<1 drinker/10 animals	15	5.57 ± 3.28	
Cleanliness			
<20% of dirty animals	25	5.65 ± 2.72	ns
≥20% of dirty animals	9	4.32 ± 2.23	
Ventilation alarm			
Absent	19	5.43 ± 2.98	ns
present	15	5.13 ± 2.21	
Access to pasture			
No	7	4.62 ± 2.17	na
Yes	26	5.61 ± 2.72	
DCT			
Blanket	5	7.85 ± 4.20	na
Selective	29	4.86 ± 2.07	
Somatic cell count			
>150,000 cells/mL	21	5.66 ± 2.84	ns
≤150,000 cells/mL	13	4.72 ± 2.26	
Cubicles material			
Other	10	6.92 ± 3.17	0.05
Straw/sawdust	23	4.74 ± 2.07	
Microbiological tests for mastitis			
Absent	3	4.04 ± 2.15	
Not on a routine basis	3	6.03 ± 0.61	na
For all problematic cows	28	5.36 ± 2.81	

sd: standard deviation; na: not applicable; ns: not significant; DCT: dry cow therapy.

## Data Availability

The data presented in this study are available on request from the corresponding author. The data are not publicly available due to privacy reason.

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
