# Peer review of "Antibiotic Use in Alpine Dairy Farms and Its Relation to Biosecurity and Animal Welfare"

_antibiotics, 2022, doi:10.3390/antibiotics11020231_

Round 1

Reviewer 1 Report

A publication on a subject of importance at the present time, however, in my opinion, it is necessary to expand the content.
In my opinion, it would be a good solution to test not only the use of antibiotics but also the development of prophylaxis against mastitis in these farms. Combining biosecurity principles with the use of antibiotics would bring tangible benefits so that growers would see a specific example that had a beneficial effect on this.

Author Response

Does the introduction provide sufficient background and include all relevant references?

Can be improved

Authors: additional relevant references have been included and some aspects have been clarified, such as the reasons behind AMU (lines 29-30) and the purpose of the National Action Plan on AMR (lines 38-39).

Is the research design appropriate?

Can be improved

Authors: the distinction between blanket dry-cow therapy and selective dry-cow therapy has been included (line 216), as well as information on prophylaxis against mastitis (Table 3).

Are the methods adequately described?

yes

Are the results clearly presented?

yes

Are the conclusions supported by the results?

Can be improved

Authors: the conclusions have been enhanced with all the study’s findings.

Comments and Suggestions for Authors:

A publication on a subject of importance at the present time, however, in my opinion, it is necessary to expand the content.

In my opinion, it would be a good solution to test not only the use of antibiotics but also the development of prophylaxis against mastitis in these farms. Combining biosecurity principles with the use of antibiotics would bring tangible benefits so that growers would see a specific example that had a beneficial effect on this.

Authors: we are thankful for the observation and we agree with the importance of including data on prophylaxis against mastitis. We were able to retrieve data on dry cow therapies in the inspected farms (blanket vs selective therapy), now reported in Tab. 3. Moreover, we added data on laboratory analyses for mastitis as follows: a. absent (in 3/34 farms); b. not on a routine basis, on pathological material (in 3/34 farms); c. for all problematic cows (in 28/34 farms). The statistical analysis was not applicable to the present sample but we added descriptive statistics in Tab. 3.

Reviewer 2 Report

Dear author,

Thanks for a well written and interesting manuscript. Your results add important information to the understanding of the comparably high AMU in Italy. For even more interesting results, it would have been interesting to include also HPCIAs in the risk factor analysis. Would that be possible to do? Apart from that, I only have some minor comments:

L 14: Maybe these abbreviations should be explained already here?

L 20: Effect or association?

L 140: DCT is applied both to cure and prevent mastitis (not only clinical) both during the dry period but mainly in the following lactation. Please consider elaborating.

L 141: Did all farms practiced blanket DCT? Or did some also use selective DCT?

L 163: animal instead of Animal.

Author Response

English language and style are fine/minor spell check required

Authors: English has been reviewed.

Does the introduction provide sufficient background and include all relevant references?

yes

Is the research design appropriate?

yes

Are the methods adequately described?

yes

Are the results clearly presented?

yes

Are the conclusions supported by the results?

yes

Comments and Suggestions for Authors

Dear author,

Thanks for a well written and interesting manuscript. Your results add important information to the understanding of the comparably high AMU in Italy. For even more interesting results, it would have been interesting to include also HPCIAs in the risk factor analysis. Would that be possible to do? Apart from that, I only have some minor comments.

Authors: We are thankful for the comments, it is a very interesting point. We actually considered it, however, given the massive reduction in 2020 of HPCIAs (only 18 farms that used at least once) and the consequent sample size reduction after stratifying by the management variables, the robustness of the statistical analysis would not have been satisfactory. We added this information at lines 262-265.

L 14: Maybe these abbreviations should be explained already here?

Authors: the abbreviations have been explained (lines 15-16).

L 20: Effect or association?

Authors: “association” is more appropriate and it replaced “effect” in the text (line 22).

L 140: DCT is applied both to cure and prevent mastitis (not only clinical) both during the dry period but mainly in the following lactation. Please consider elaborating.

Authors: we agree with the importance of including data on dry-cow therapy. We were able to retrieve data on dry cow therapies in the inspected farms. More specifically, we distinguished between selective dry-cow therapy and blanket dry-cow therapy (Table 3, lines 72-74, 152-153, 216-217); however, in the case of blanket dry-cow-therapy, it was not possible to discriminate between treatments applied either to cure or prevent mastitis.

L 141: Did all farms practiced blanket DCT? Or did some also use selective DCT?

Authors: we added this information in Table 3 and lines 72-74, 152-153, 216-217, and 29 out of 34 farms applied selective DCT.

L 163: animal instead of Animal.

Authors: the capital letter has been replaced with the lower case letter (line 179).

Reviewer 3 Report

The manuscript is well written and can be published in ANTIBIOTICS. Some minor correction should be made.

Line

Comment

5

all are 1, so it can be omitted

28

Is it really necessary? Explain in more detail when it is necessary: Crowding, to early moving of calves, etc.

36

Is this really the ultimate goal? Explain in more detail.

55

Is the p-value for the whole decrease or only for the difference between 2020 and 2018?

100

ESVAC? explain

114

PCU: explain?

162

as well as

179

Give more information, i.e. altitude, alpine region, and so on.

207

explain EMA (for readers of non EU countries): European Medicines Agency

323

… fill in

326

… fill in

Author Response

Moderate English changes required

Authors: English has been reviewed.

Does the introduction provide sufficient background and include all relevant references?

yes

Is the research design appropriate?

yes

Are the methods adequately described?

yes

Are the results clearly presented?

yes

Are the conclusions supported by the results?

Yes

Comments and Suggestions for Authors

The manuscript is well written and can be published in ANTIBIOTICS. Some minor correction should be made.

Authors: we are thankful for your comments. We answered your observation as follows.

Line - Comment

5- all are 1, so it can be omitted

Authors: Number 1 has been omitted (line 4-6).

28 - Is it really necessary? Explain in more detail when it is necessary: Crowding, the early moving of calves, etc.

Authors: we agree that the use of antibiotics is necessary to avoid animals suffering from bacterial infections and therefore to guarantee animal health and welfare (line 29-30).

36 - Is this really the ultimate goal? Explain in more detail.

Authors: as specified in lines 38-39, the ultimate goal is to provide a coordinated and sustainable strategy to address AMR nationwide.

55 - Is the p-value for the whole decrease or only for the difference between 2020 and 2018?

Authors: The p-value is for the whole decrease (comparison 2020 vs 2018 and 2020 vs 2019). Hopefully, it will be clearer by editing the sentence as follows: “with a significant decrease (P<0.001) in 2020 compared with both 2018 (-20.1%) and 2019 (-21.3%)” (line 59-60).

100 - ESVAC? Explain

Authors: the abbreviation has been explained: ESVAC is the abbreviation of “European Surveillance of Veterinary Antimicrobial Consumption” (line 108).

114 - PCU: explain?

Authors: the abbreviation has been correctly explained: PCU stands for “Population Correction Unit” (line 123).

162 - as well as

Authors: “and” has been replaced by “as well as” (line 177)

179 - Give more information, i.e. altitude, alpine region, and so on.

Authors: we are thankful for the observation. We added more information at lines 194-198 to better characterize the farms: Thirty-four free-stall farms of mixed-breed dairy cows (i.e. Holstein Friesian, Brown Swiss, Simmental cows) in the Province of Trento (North-Eastern Italy) were randomly selected for this study. Farms were located in the area between Garda lake (south), Stelvio Natural Park (west), Pale di San Martino Natural Park (east) and Bolzano (north), and the altitude ranged from 400 m to 1700 m.

207 - explain EMA (for readers of non EU countries): European Medicines Agency

Authors: the abbreviation has been explained (line 228).

323 - … fill in

Authors: the missing information has been reported (line 371).

326 - … fill in

Authors: the missing information has been reported (line 374)